# Poly(Alkylene 2,5-Thiophenedicarboxylate) Polyesters: A New Class of Bio-Based High-Performance Polymers for Sustainable Packaging

**DOI:** 10.3390/polym13152460

**Published:** 2021-07-27

**Authors:** Giulia Guidotti, Michelina Soccio, Massimo Gazzano, Valentina Siracusa, Nadia Lotti

**Affiliations:** 1Department of Civil, Chemical, Environmental and Materials Engineering, University of Bologna, Via Terracini 28, 40131 Bologna, Italy; giulia.guidotti9@unibo.it; 2Interdepartmental Center for Industrial Research on Advanced Applications in Mechanical Engineering and Materials Technology, CIRI-MAM, University of Bologna, 40126 Bologna, Italy; 3Institute of Organic Synthesis and Photoreactivity, ISOF-CNR, Via Gobetti 101, 40129 Bologna, Italy; massimo.gazzano@isof.cnr.it; 4Department of Chemical Science, University of Catania, Viale A. Doria 6, 95125 Catania, Italy; vsiracus@dmfci.unict.it; 5Interdepartmental Center for Agro-Food Research, CIRI-AGRO, University of Bologna, 40126 Bologna, Italy

**Keywords:** 2,5-thiophenedicarboxylic acid, thermal properties, barrier properties, mechanical properties, 2D-ordered structure, structure-property relationship

## Abstract

In the present study, 100% bio-based polyesters of 2,5-thiophenedicarboxylic acid were synthesized via two-stage melt polycondensation using glycols containing 3 to 6 methylene groups. The so-prepared samples were characterised from the molecular point of view and processed into free-standing thin films. Afterward, both the purified powders and the films were subjected to structural and thermal characterisation. In the case of thin films, mechanical response and barrier properties to O_2_ and CO_2_ were also evaluated. From the results obtained, it emerged that the length of glycolic sub-units is an effective tool to modulate the chain mobility and, in turn, the kind and amount of ordered phases developed in the samples. In addition to the usual amorphous and 3D crystalline phases, in all the samples investigated it was possible to evidence a further phase characterised by a lower degree of order (mesophase) than the crystalline one, whose amount is strictly related to the glycol sub-unit length. The relative fraction of all these phases is responsible for the different mechanical and barrier performances. Last, but not least, a comparison between thiophene-based homopolymers and their furan-based homologues was carried out.

## 1. Introduction

Plastic waste has become a matter of great concern in recent years and is pushing governments and society to come up with new strategies to face the problem. If the amount of both plastic waste volume and the exploitation of non-renewable sources are analysed, it is immediately clear how the development of sustainable alternatives to traditional plastics is urgent and re-use, where possible, should be encouraged. If it is true that since 2006, the amount of plastic waste recycled has doubled, it is also likely that we are still far from the so called “zero landfilling” needed to achieve the circular economy of plastics, as in many cases plastic waste is sent to landfills or, even worse, is disposed of in marine and terrestrial environments [1].

It is also important to consider that among the principal current applications of plastics, packaging alone covers 39.9% of them [1]: as far as packaging is concerned, petrochemical-based ones are currently largely used thanks to its abundance, great mechanical strength, and barrier properties together with low production costs despite most of it being nonbiodegradable. As an alternative, plastics obtained from renewable sources can represent a valuable solution thanks to their resource efficiency and reduced carbon footprint. This potential is confirmed also by the increasing bioplastics market size and demand: according to the most recent studies, global bioplastics production capacity is estimated to grow from 2.11 million tons in 2019 to about 2.43 million tons in 2024 [2]. However, to date there are still some important drawbacks related to the final properties of these bioplastics, limiting their current applications, such as short-term stability, bad processability, poor mechanical strength, brittleness, or relatively high gas permeability.

Besides the use of renewable sources, also reducing packaging volumes could be a very practical route to reduce environmental impact. For this reason, many efforts have been devoted to the research of optimized flexible solutions in place of rigid and oversized ones, keeping at the same time the pack quality and the product protection, in particular when the product is characterised by a very short life cycle, as in the case of food. For example, vacuum skin packs can provide excellent barrier properties, minimizing volumes while at the same time preserving the product’s shelf life. Another issue is related to multilayer materials, commonly used to better preserve food, in which high barrier properties are ensured by the combination of different layers. These strata are intimately overlapped and very thin, and thus very difficult to separate in view of a possible reuse. Therefore, the use of monomaterials, which can provide an effective barrier to gases on one side and can be also easily recovered after use, should be preferred.

Aware of the severity and the complexity of this scenario, the European Commission has adopted both a Waste Framework Directive (2008/98/EC) and an action plan for a circular economy in view of ensuring that all plastic packaging would be recyclable by 2030 [3]. In addition, the U.S. Department of Energy has identified twelve building blocks that can be produced from renewable sources, such as sugars, which can in turn be converted to many high-value bio-based chemicals [4]. Among them, 2,5-furandicarboxylic acid (FDCA) is of particular interest, both academically and industrially, since one of its derived materials, poly(ethylene furanoate) (PEF), is considered the most credible green alternative of the fossil-based poly(ethylene terephthalate) (PET), which is largely employed in packaging applications. Moreover, according to published studies, PEF is more performant than PET in terms of mechanical and barrier response [5,6,7,8,9]. It is known that FDCA can be obtained from sugar dehydration, obtaining 5 hydroxymethyl-furfural (HMF) as an intermediate product [10], with high yield and purity.

In recent years, 2,5-thiophenedicarboxylic acid (2,5-TDCA) has attracted growing attention due to its chemical structure, which is similar to that of FDCA. In addition, 2,5-TDCA can be derived from renewable sources and is already industrially produced from adipic acid, which can be obtained, in turn, from glucaric acid, muconic acid, or lignin [11,12,13,14]. As proof of the growing interest in these polyesters, many scientific studies have been recently published, including those conducted outside of our research group, demonstrating the great potential of both thiophene-based homopolymers and copolymeric systems [15,16,17,18,19,20,21,22,23,24].

Moreover, since the enzymatic degradability of PET and PEF has already been demonstrated [25,26,27], more recently the biodegradability of poly(butylene 2,5-furandicarboxylate) PBF and poly(butylene 2,5-thiophenedicarboxylate) PBTF was also investigated. It was proved that *Humicola insolens* (HiC) and *Thermobifida cellulosilytica* (Cut) cutinases can hydrolyse both homopolymers, opening a new way for the industrial recycling of furan- and thiophene-based materials [28].

Considering the above-described scenario, the present work aimed to synthesize by two-step polycondensation four fully bio-based poly(alkylene 2,5-thiophenedicarboxylate)s, starting from dimethyl 2,5-thiophenedicarboxylate, the dimethyl ester of 2,5-TDCA, and glycols of different length (the number of methylene groups changed from 3 to 6). These can be all derived from renewable sources as well as 2,5-TDCA. By acting only on the number of CH_2_ groups in the glycol moiety, it was possible to cover a wide range of properties, making the materials obtained particularly suitable for the realisation of rigid as well as flexible films for packaging applications. The polymers synthesized, in the form of both powders and thin films, were characterised from the molecular, structural, thermal, mechanical, and gas barrier point of view, and correlations between structure and properties were extrapolated. In addition, the presence of a mesophase was hypothesized, and both the amount of ordered domains developed and the mechanism of their formation was compared to those of furan-based analogues [29].

## 2. Materials and Methods

### 2.1. Materials

2,5-thiophenedicarboxylic acid (2,5-TDCA) was purchased from TCI (Tokyo, Japan), titanium tetrabutoxide (TBT), titanium isopropoxide (TIP), 1,3-propanediol (PD), 1,4-butanediol (BD), 1,5-pentanediol (PeD), and 1,6-hexanediol (HD) were purchased from Sigma-Aldrich (Milan, Italy). All reagents were used as received.

### 2.2. Synthesis

Prior to polymer synthesis, dimethyl 2,5-thiophenedicarboxylate (DMTF) was prepared, starting from 2,5-TDCA, according to the procedure described in [18]. Afterward, all the homopolymers were synthetized through two-stage melt polycondensation, starting from DMTF (0.02 mol, 4 g) and PD/BD/PeD/HD (0.04 mol, corresponding to 3.04 g for PD, 3.60 g for BD, 4.17 g for PeD and 4.73 g for HD, respectively), using TBT and TTIP as catalysts (200 ppm of each). In all cases, a diester:glycol molar ratio of 1:2 was used in order to favour the DMTF solubilisation. The syntheses were carried out in a glass reactor put in a termostated bath. The reaction mixture was kept under continuous stirring by using a two-bladed centrifugal stirrer connected to an overhead motor. Syntheses proceeded in two stages. The first one occurred under pure nitrogen flow at a temperature of 180 °C. This condition was maintained until 90% of the theoretical amount of methanol was distilled off (about 2 h from the solubilisation of the dimethyl ester). During the second stage, pressure was reduced gradually to 0.06 mbar and temperature raised up to 220 °C to favour the removal of the glycolic excess as well as the increasing of molecular weight. Polymerisation was stopped when a constant torque value was reached (after about two additional hours).

Prior to solid-state polymer characterisation, all the samples were purified to remove impurities, oligomers, and catalysts. The polyesters were first dissolved in chloroform (in the case of PPTF some drops of hexafluoro-2-propanol were needed for the complete polymer solubilisation), then precipitated in methanol. Lastly, the purified powders were kept under a vacuum at room temperature for 48 h to remove the residual solvent.

### 2.3. Molecular Characterisation

Chemical structure was confirmed by means of proton nuclear magnetic resonance (^1^H-NMR), employing a Varian Inova 400 MHz instrument (Palo Alto, CA, USA). More in detail, each homopolymer was dissolved in deuterated chloroform containing 0.03 vol.% tetramethylsilane (TMS) as an internal standard (solutions’ concentration was about 0.5 wt%). For PPTF, a mixture of trifluoracetic acid and chloroform was used (20% *v/v*). The spectra were recorded at room temperature with a relaxation delay of 1 s, acquisition time of 1 s and up to 64 repetitions.

In order to determine the molecular weight of the synthesized materials, gel permeation chromatography (GPC) was performed at 30 °C using an 1100 HPLC system (Agilent Technologies, Santa Clara, CA, USA) equipped with a PLgel 5-mm MiniMIX-C column. A UV detector was employed, and chloroform was used as an eluent. We used 0.3 mL/min flow and sample concentrations of about 2 mg/mL. Polystyrene standards in the range of 800–100,000 g/mol were used to obtain a calibration curve.

### 2.4. Film by Compression Moulding

Polymeric films of about 100 μm thickness were prepared via compression moulding using a Carver laboratory press (Wabash, IN, USA). The PPTF, PBTF, PPeTF and PHTF samples were melted at temperatures of 205, 180, 95 and 125 °C, respectively, between two Teflon sheets. Therefore, a pressure of 7 ton·m^−2^ was applied for 2 min. The samples were then ballistically cooled down to room temperature in press, the process taking about 15 min. The as-obtained film pictures are shown in Appendix A. Prior to further tests, the films of the polymers with a glass transition temperature below room temperature (PPeTF and PHTF) were stored at room temperature for 3 weeks in order to let them reach equilibrium crystallinity.

### 2.5. Thermal and Structural Characterisation

Thermal stability and thermal transitions of the homopolymers were investigated by means of thermogravimetric analysis (TGA) and calorimetric measurements (DSC), respectively. In the former case, the analyses were carried out using a Perkin Elmer TGA7 apparatus (gas flow: 40 mL/min), under an inert N_2_ atmosphere (Shelton, CT, USA). Weighed samples of about 10 mg were heated at a constant rate (10 °C/min) from 40 °C up to 800 °C. In this way, it was possible to calculate the temperature corresponding to initial degradation (T_onset_) as well as the temperature at which maximum weight loss occurs (T_max_).

As for calorimetric measurements, these ones were carried out on both purified powders and films using a Perkin Elmer DSC6 (Shelton, CT, USA) calibrated with indium and cyclohexane standards. As is known, DSC analysis allows the determination of glass transition temperature (T_g_), melting and crystallisation temperatures (T_m_ and T_cc_, respectively), as well as the specific heat increment associated with the glass transition of the amorphous phase (Δc_p_), and the heat of fusion and crystallisation (ΔH_m_ and ΔH_cc_, respectively), relative to the crystalline portion of the polymer. T_g_ was calculated as the midpoint of the heat capacity increment related to the glass-to-rubber transition. T_m_ and T_cc_ were considered as the peak values of the endothermal and exothermal phenomena in the DSC curve, respectively, Δc_p_ was calculated from the vertical distance between the two extrapolated baselines at T_g_, while ΔH_m_ and ΔH_cc_ were calculated considering the global area subtended by melting and crystallisation peaks, respectively. In the procedure adopted, polymeric samples of about 8 mg were put in aluminium pans and heated at a constant rate (20 °C/min) from −50 °C to 210, 200, 160 and 180 °C for PPTF, PBTF, PPeTF and PHTF, respectively, held there for 3 min, and then rapidly cooled (100 °C/min) to −50 °C. They were then heated again at the same heating rate (II scan).

Wide-angle X-ray scattering (WAXS) patterns of purified powders and films were recorded by means of a PANalytical X’PertPro diffractometer (Almelo, The Netherlands) equipped with a fast solid-state X’Celerator detector and a copper target (λ = 0.1548 nm). The samples were scanned in the range 2θ = 5° to 60° (step size of 0.10°, acquisition time of 100 s per step). The degree of crystallinity (X_c_) was calculated dividing the value of the crystalline diffraction area (A_c_) by the total area of the diffraction pattern (A_t_), X_c_ = A_c_/A_t_. A_c_ was determined by subtracting the amorphous halo from the A_t_ value itself.

### 2.6. Mechanical Characterisation

Tensile tests were carried out on rectangular films (5 mm × 50 mm, gauge length of 20 mm) by means of an Instron 5966 testing machine (Norwood, MA, USA) equipped with a rubber grip and a 1KN load cell. Experiments were performed at room temperature with a crosshead speed of 10 mm/min, testing at least seven specimens for each polymeric sample. The so-obtained load-displacement curves were then converted to stress–strain curves. The value of the tensile elastic modulus (E) was calculated from the initial linear slope of the stress–strain curve. The values of stress at break (σ_B_) and elongation at break (ε_B_) were also determined. The results are reported as the average value ± standard deviation.

### 2.7. Gas Barrier Properties Evaluation

In order to evaluate gas transmission rates through polymeric films, permeability tests were performed by using a manometric method (Permeance Testing Device, type GDP-C (Brugger Feinmechanik GmbH, München, Germany)), in accordance with ASTM 1434-82, DIN 53 536, ISO/DIS 15 105-1 and the gas permeability testing manual of the instrument. Each circular film (film area of 78.5 cm^2^) was placed in between two chambers. First, the system was put under high vacuum, and then the upper chamber was filled with the selected gas (CO_2_ or O_2_ food grade, 0% RH) at 23 °C and atmospheric pressure, with a gas stream of 100 cm^3^/min. Any variation in gas pressure was recorded, as a function of time, by a pressure sensor located in the lower chamber. Gas transmission rate (GTR [cm^3^ cm m^−2^ d^−1^ atm^−1^]), i.e., the value of film permeability, was determined taking into account the pressure increase as a function of time and of the whole volume of the device. The GTR values were normalised for the film thickness, the latter measured as the average value among five different measurements.

## 3. Results and Discussion

### 3.1. Synthesis and Molecular Characterisation

The chemical structure of the homopolyesters under investigation are reported in Figure 1, together with a schematic representation of the synthetic process. All the materials were characterised by a diacid aromatic subunit derived from 2,5-TDCA and by an aliphatic glycolic subunit differing for the number of methylene groups ranging from 3 to 6. It is worth noting from the chemical structure a nonpolar moiety, coming from the aliphatic segment, was combined with a polar one, i.e., thiophene ring, whose dipole moment (μ) was 0.51 D. The starting 2,5-TDCA was esterified because the impurities present in the diacid acting as nucleating agents affected the polymer microstructure and consequently the final polymer properties, such as the mechanical and gas barrier ones [18]. The as-synthesized polymers looked slightly coloured due to the presence of residual catalysts, while after purification, in all cases, white powders were obtained.

The chemical structures were confirmed via ^1^H-NMR spectroscopy (Figure 2). Table 1 collects the peak position and multiplicity observed in the NMR spectra. No extra signals were found, proving the absence of impurities in the synthesized samples.

All the materials analysed show high and comparable values of M_n_ (see Table 1 and the chromatograms shown in Appendix A) and a polydispersity index (D) around 2, typical of polyesters obtained by melt polycondensation, corroborating the optimization of the polymerisation process.

### 3.2. Thermal Characterisation

#### 3.2.1. Powder Samples

The first scan DSC curves of powder samples are shown in Figure 3A and the relative data are displayed in Appendix A. The DSC traces of all samples present the typical profile of semicrystalline materials with an endothermic baseline deviation related to glass-to-rubber transition, followed by an endothermic phenomenon associated with the melting of crystalline phase. This phase behaviour can be explained due to solvent-induced crystallisation. In PPTF DSC trace, the glass transition phenomenon is not detectable because of its very low intensity due to the high crystallinity degree. Besides the main melting peak, two low and broad endothermic signals can be identified around 60–70 °C, whose origin is not so clear. Broad multiple melting peaks were also detected for PBTF, PPeTF and PHTF samples, suggesting melting–crystallisation–re-melting processes occurring during heating scan or in the presence of different crystalline phases.

In general, the position of the main endotherm (T_m_) moved toward lower temperature as the glycol length increased, with the exception of PPeTF, which exhibited the lowest melting temperature.

As is well known, semicrystalline polymers behave differently from their completely amorphous analogues due to the physical crosslinks provided by crystalline structures, which limit chain mobility, raising the T_g_ value [30,31]. Therefore, we decided to perform a DSC scan after rapid cooling from the molten state with the aim of quenching the polymer macromolecules. The relative curves are reported in Figure 3B, while the corresponding calorimetric data are displayed in Appendix A. The results demonstrated the effectiveness of the rapid cooling from the melt in quenching the powders, being all the materials amorphous in the II scan. Nevertheless, some differences in the crystallisation capability from the glass could be evidenced. In particular, in the second run PPeTF and PHTF showed just the endothermic jump associated with the glass-to-rubber transition, while PPTF and PBTF powders in addition to the T_g_ step presented an endothermic signal together with an exothermic one of equal intensity, demonstrating their chains are capable of rearranging into an ordered structure upon heating. However, since ∆H_c_ ≈ ∆H_m_, one can assert rapid cooling from the melt was effective in quenching both PPTF and PBTF powders. Considering the T_g_ values, as expected, the higher the number of methylene groups the higher the flexibility of macromolecular chains, i.e., the lower the T_g_ value. Moreover, a longer glycol subunit implies fewer stiff aromatic rings along the polymer backbone. More in detail, PPTF was the only glassy material at room temperature, being its T_g_ > RT. Conversely, PPeTF and PHTF were in the rubbery state, with T_g_ well below room temperature. PBTF was instead characterised by a glass transition temperature near room temperature.

WAXS patterns of powder samples are reported in Figure 3C. At first glance, it is clear that the samples were characterised by different crystallinity degrees, X_c_ being equal to 29%, 16%, 9% and 25% for PPTF, PBTF, PPeTF and PHTF, respectively. The profiles showed different shapes with a variation of peak positions on the 2-theta scale and intensities peculiar for each sample. This suggests the crystal phases probably do not share any similarities, i.e., are not isomorphous. By comparing with previous reported data [15], we can ascribe the crystal fraction of the PPTF sample to its β-form. Indeed, the four main peaks at 9.3°, 16.3°, 23.3° and 25.5° were previously reported for a film sample named β-PPTF [15]. As for the PBTF sample, the observed main peaks at 8.2°, 14.0°, 16.2° and 24.8° permitted us to recognize in the PBTF powder the γ-phase [18]. PPeTF showed a very low crystallinity degree with peaks, or better peak clues, at 7.6°, 16.2°, 21.6°, 22.8°, 24.3° and 31.1°. The PHTF profile was characterised by well-defined peaks identified at 6.5°, 19.2°, 20.0°, 22.6°, 24.9°, 28.6° and 31.3°, indicating the presence of a conspicuous amount of crystals. This pattern is compatible with the one reported by authors of another study [19].

#### 3.2.2. Compression-Moulded Films

Prior to solid-state characterisation, compression-moulded films were subjected to 3 weeks storage at room temperature to uniform their thermal history since both PPeTF and PHTF are characterised by T_g_s below room temperature, and the T_g_ of PBTF is near RT.

Firstly, compression-moulded films (Appendix A) were subjected to thermogravimetric analysis (TGA) under pure nitrogen flux. The temperatures of initial degradation (T_onset_) and of maximum weight loss rate (T_max_) are displayed in Table 2, while the corresponding TGA curves and derivatives are shown in Figure 4A,B, respectively. All the polyesters displayed very good thermal stability, with T_onset_ above 370 °C and T_max_ above 400 °C, as clearly evidenced by thermograms’ derivatives, the degradation process occurring in all cases in two steps, the main one around 400 °C. The second step was sensibly smaller in all cases and occurred immediately afterward. Lastly, no residual mass at 800 °C was found. Some differences among the samples could be found. PPTF turned out to be the fastest-degrading material, presumably because of the high amount of ester groups, which undergo thermal cleavage more easily, and the presence of 1,3-propanediol, which favours β-scission reactions as previously observed for other polyesters, among these PPF, its furan-based counterpart [29,32]. PBTF was, on the contrary, the most thermally stable among the family, while PPeTF and PHTF were characterised by a very similar thermal stability, intermediate between those of PPTF and PBTF. This trend was slightly different from the one observed in the case of furan-based materials, where PPeF was the most thermally stable [33].

Each thiophene-based polyester was directly compared with its furan-based counterpart (see Figure 4C). As you can see, PPTF and PBTF appeared to be more thermally stable than their furan-based counterparts (PPF and PBF). We hypothesize such a result could be due to the higher resonance energy, p-to-d π-back bonding, and lack of ring strain due to the longer C–S bond of the thiophene ring. Conversely, poly(pentamethylene furanoate) (PPeF) turned out to be more thermally stable than PPeTF, whereas PHF and PHTF showed very similar thermal stability.

The highest thermal stability of PPeF within the furan-based family was ascribed to the presence in this polyester of the highest fraction of mesophase, the latter due to the establishment of hydrogen bonds between adjacent macromolecular chains in addition to π–π interactions [33]. As previously reported, mesophase formation is favoured in flexible and mobile macromolecular chains and compete with the crystalline 3D phase. PPeF was indeed amorphous with T_g_ below room temperature and had very slow crystallising capability [29,33,34]. The authors of the present paper previously investigated PBTF, which was characterised by the presence of a 2D-ordered phase (mesophase) [16,18]. Nevertheless, the mesophase present in thiophene-based polyesters cannot originate from hydrogen bonds between adjacent polymer chains, sulphur atoms being significantly less electronegative than oxygen. In this case, the 2D-ordered structure could arise from thiophene ring π–π stacking, as observed also for polythiophenes [35] and for some other aromatic polyesters containing, for example, terephthalic and isophthalic rings [36,37,38]. As previously reported [16,18], also for the thiophene-based family, the mesophase formed at the expense of the crystalline one, particularly enhanced by compression-moulding treatment.

The DSC analysis discussed below indicates PBTF compression-moulded film is characterised by a very low crystallinity degree, different from the other members of the family. Consequently, we can hypothesize the mesophase fraction, and then the interchain π–π stacking interactions, reached the maximum value in the PBTF film, determining the highest thermal degradation temperature.

The first scan DSC traces of compression-moulded films are shown in Figure 5 and the relative data are displayed in Table 2.

First of all, it has to be emphasized that some remarkable differences in the thermal behaviour before and after compression moulding were noticed. In all cases, with the exception of PBTF film, the I scan curves show endotherms typical of semicrystalline materials. In the case of PBTF, an additional exothermic peak between T_g_ and T_m_ could be observed; however, the peak areas were comparable, (∆H_cc_ = ∆H_m_), indicating film processing totally suppressed the development of 3D-crystalline phase in the PBTF sample. As for PPeTF, analogously to powder, a broad endothermic phenomenon could be found. However, in this case, the peak at the lower temperature (52 °C) was more intense than that at the higher temperature (63 °C). Finally, the DSC curve of PHTF film shows two melting peaks, at 49 and 95 °C, very similar to the endotherms measured for purified powder, but again higher intensity for the lower temperature signal was detected. Interestingly, in both PPeTF and PHTF polyesters the first endothermic peak was located at a constant temperature, about 50 °C. It is worth noticing the signal, similar in terms of position even if less intense, previously reported for PBTF film was ascribed to the mesophase isotropisation phenomenon [16]. Such an endotherm is not visible in the calorimetric trace of the PBTF film obtained from dimethyl 2,5-thiophenedicarboxylate [18], probably hidden by the upcoming cold crystallisation peak (as known, the mesophase can act as a precursor of the 3D crystalline phase). Considering this scenario, it can be hypothesized that the endotherms around 50 °C in both PPeTF and PHTF were due to this peculiar 2D microstructure already detected and investigated.

As expected, II scan DSC curves of films are identical to those collected for purified powders.

In Figure 5C, the WAXS profiles of films are reported. From the profile shape, we can state PPTF and PHTF were in the same crystal forms as the respective powders. Indeed, the peak positions of PPTF film were 16.5°, 23.3° and 25.5°, very similar to the powder ones; PHTF showed peaks at 6.5°, 19.1°, 20.0°, 22.2°, 24.4°, 28.2° and 31.2°. For this sample, the position of the peaks was not exactly the same as those observed in the powder sample, but the good crystallinity and the overall match of the pattern allow us to state the powder and the film displayed the same crystal form, the small differences observed being due to different preparations. PBTF film showed only the bell-shaped background, typical of amorphous compounds, in agreement with DSC results. The comparison of PPeTF samples was not easy, the film showing small peaks or shoulders at 7.5°, 16.3°, 18.4°, 21.3° 22.6° and 24.3°. They were roughly at the same positions found in the powder sample. The imperfect coincidence in the position of the peaks as well as the presence of additional peaks could be compatible with the presence of a further phase.

As for sample crystallinity, X_c_ values were 18%, 14% and 21% for PPTF, PPeTF and PHTF, respectively.

For the sake of comparison, in Appendix A T_g_ and T_m_ of the thiophene-based polyesters under study and their furan- and terephthalate-based counterparts are reported as a function of glycolic subunit length. As for T_g_ values, regardless of the kind of ring considered, all the families showed a regular decrease of glass transition temperature with the glycolic subunit length. In addition, by fixing the glycol length, the trend observed was T_g,thiophene_ < T_g,benzene_ ≤ T_g,furan_, more pronounced for 1,3-propanediol- and 1,4-butanediol-containing polymers, as a result of the different polarity/aromaticity ratio of the three rings. The lower T_g_ values of thiophene-based polyesters with respect to those of the furan-based ones can be explained considering that in the former polymer family interchain interactions are weaker because of the lower electronegativity of sulphur atoms with respect to oxygen ones.

As for the melting temperatures, although the same trend can be noticed for all the polymeric families, the decrease of T_m_ for thiophene-based samples was shifted at higher CH_2_ numbers with respect to their furan- and benzene-based analogues, probably due to differences in the main structural parameters, which led to a less efficient chain packing, i.e., lower T_m_s. PPTF and PPF were exceptions: the higher T_m_ of the former polymer can be ascribed to a more perfect crystalline phase as well as to the higher aromaticity of thiophene ring, which promotes chain folding [15].

### 3.3. Mechanical Characterisation

Mechanical properties of the compression-moulded films were evaluated by tensile tests. The values of the elastic modulus (E), stress at break (σ_B_), and strain at break (ε_B_) are displayed in Table 3. The corresponding stress–strain curves are shown in Figure 6.

The results obtained can be explained on the basis of chain flexibility (i.e., T_g_ value) and crystallinity degree, which are the main parameters affecting polymers’ mechanical response. More in detail, referring to mechanical characterisation data obtained after three weeks of storage at room temperature, it was possible to notice that PPTF was the most rigid material, with the highest value of E and the lowest elongation at break. This result was due both to its crystallinity degree, the highest among the family, and its T_g_, the latter being well-above room temperature, which limited macromolecular mobility. Among the thiophene-based family, PBTF, which was amorphous with T_g_ around RT, turned out to be the one with the lowest value of E and the highest of ε_B_, almost 600%. Regarding polymers containing longer and flexible glycols, i.e., PPeTF and PHTF, which are both semicrystalline and in the rubbery state at RT, they were characterised by a mechanical response intermediate between those of PPTF and PBTF. The elongation at break was similar for the two polymers, whereas PPeTF exhibited a higher elastic modulus despite its lower crystallinity degree probably because of its lower chain flexibility (higher T_g_). Lastly, for all the materials analysed, stress at break was comparable, with values ranging from 13 to 19 MPa.

The unexpected mechanical response of PTBF can be explained as being due to the presence of a mesophase in the polymeric film, as already demonstrated in previous studies carried out by some of the authors of the present paper [16,18]. As reported in the literature, polymer liquid crystals are characterised by smart mechanical properties, their brittleness being significantly reduced (high values of elongation at break).

In order to support the already-mentioned hypothesis of the presence of a mesophase, stress–strain measurements were carried out on PPeTF and PHTF films immediately after compression moulding, as reported in Figure 7. In these conditions, the two films were completely amorphous, as evidenced by DSC (only the endothermic baseline deviation associated with glass transition was evident in the DSC trace of both polyesters) and WAXS analysis (WAXS profiles of both polymers show only the bell shape halo characteristic of the amorphous phase) (Figure 7B,C,E,F). Nevertheless, it is worth noting that although both materials were in an amorphous rubbery state at room temperature, free-standing films could be obtained. That supports the assumption of the presence, together with the amorphous fraction, of another phase. A similar behaviour was already evidenced for PPeF, the furan-based analogous of PPeTF [29,33], for which the presence of a mesophase was demonstrated. As one can see from Figure 7, the mechanical response of the films tested immediately after moulding was completely different. Both films were characterised by outstanding elongation at break (around 1700% in the case of PPeTF and about 700% for PHTF) and elastic modulus values, about two orders of magnitude lower than those of the same films measured after 3 weeks from moulding (Figure 7A,D). This result further supposes the development of a mesophase in PPeTF and PHTF, too. The storage of the films at room temperature also favoured, of course, the formation of a 3D-crystalline phase. The former is associated with the endothermic phenomenon at constant T around 50 °C, the latter to the fusion process occurring at a higher temperature whose value increases with the number of methylene groups present in the glycol subunit (see Figure 7B,E). Both the films become more rigid (E increases) and stretch much less after 3 weeks at room temperature due to the increment of crystalline fractions and to disclinations between amorphous, 2D and 3D ordered phases.

In Figure 8, a comparison between mechanical characterisation data of thiophene-based homopolymers and their furan-based counterparts is shown. As far as the elastic modulus is concerned, an odd–even -CH_2_- number trend was observed: the thiophene-based homopolymers containing a glycol subunit with an odd number of -CH_2_- groups (PPTF and PPeTF) showed a higher elastic modulus with respect to their furan-based counterparts, attributable to their semicrystalline nature. An opposite result was found for the samples containing an even number of C atoms in glycolic subunits (PBTF vs. PBF and PHTF vs. PHF); in fact, the thiophene-based polyesters showed a lower elastic modulus than the furan-based analogous. More in detail, PBF film was semicrystalline, whereas PBTF was fully amorphous and PHF had a higher fraction of crystalline phase compared to PHTF. Moreover, the PHF crystal phase was characterised by a higher melting temperature with respect to PHTF, indicating a higher degree of perfection, i.e., the crystal phase was highly packed.

In the case of stress at break, we measured in general a lower value for thiophene-based polyesters when compared with their furan-based counterparts, with the exception of polyesters of 1,5 pentanediol. For PPeTF and PPeF, exactly the opposite occurred. Interestingly, σ_B,PPTF_ < σ_B,PPF_, despite the semicrystalline nature of the former film and the amorphous nature of the latter one. Such a result can be correlated with the significantly higher T_g_ of PPF. For the pair PBTF and PBF, the lower σB of the thiophene-based polyester was due to its amorphous nature, PBF being on the contrary semicrystalline. As far as PPeTF and PPeF are concerned, σ_B,PPeTF_ > σ_B,PPeF_ because of the semicrystalline nature of thiophene-containing film and the amorphous character of the furan-based one. Lastly, PHF film is characterised by a higher stress at break than the PHTF one due to its higher crystalline degree, T_m_ and T_g_ values.

The elongation at break was also correlated with the presence in the film of a crystalline phase; in fact, ε_B,PBTF_ > ε_B,PBF_, (PBTF film was amorphous, while PBF contained a crystal phase). Exactly the opposite happened for the PPeTF and PPeF pair; the former was indeed semicrystalline and the latter was amorphous. Lastly, PHTF and PHF were characterised by a very similar elongation at break, as expected if we considered both polyesters are semicrystalline and in the rubbery state.

### 3.4. Gas Barrier Properties Evaluation

Barrier properties were evaluated with respect to dry O_2_ and CO_2_ gases. The GTR values normalised for the sample thickness and measured at 23 °C and at 0% of relative humidity are displayed in Table 3. For the sake of comparison, in Figure 9A,B gas permeability values of thiophene-based materials are shown together with those of their furan-based counterparts and with some commonly used traditional fossil-based plastics and other bio-based polymers, respectively.

As is well-known, many factors, such as ordered and amorphous phases amount, glass transition temperature, chain polarity and flexibility, as well as molecular weight and its distribution, can affect the final barrier properties of materials together with the different characteristics of gas molecules (like size, polarity, inertness). Since gases cannot diffuse and permeate through the highly packed crystalline phase, this parameter plays a key role. Therefore, it is generally assumed that in polymers with higher percentages of crystalline phases the best barrier performances are observed. In addition, it is well-recognized the glassy state offers a higher barrier to gases since it is characterised by reduced chain mobility with respect to the rubbery one, and also by a lower free volume through which gas can diffuse. When mesogenic groups, like thiophene rings, are present together with flexible segments, barrier properties can further improve thanks to the development of a mesophase (1D- or 2D-ordered structure), as this is even more effective than the crystalline phase in hampering gas passage [39]. As already mentioned, meso- and crystalline phases develop one at the expense of the other. Moreover, when both form the amorphous–mesophase–crystal disclination content raises. According to these assumptions, it is not surprising PBTF turned out to be the best-performing material of the family, being the sample in which just the most performant mesophase was present together with the disordered portion, i.e., less interphase percentage.

In addition, previous studies carried out on poly(butylene isophthalate) PBI, which similar to PBTF is aromatic with a T_g_ near T_room_, have shown this material, if stored at room temperature, develops a very efficiently packed amorphous phase within a few minutes [40,41]. The presence of this peculiar dense amorphous phase can explain the excellent mechanical and barrier properties of PBI. Conversely, in poly(ethylene terephthalate) (PET), another well-known aromatic polyester, a very different behaviour was observed: being PET T_g_ is well-above room temperature, it is not capable of quick re-arrangement, and the above-mentioned densification does not occur in short times. All these studies then seem to support the assumption according to which the mesophase presence and the possibility of quickly compacting its mobile amorphous phase are responsible for the outstanding PBTF properties. According to the data reported in Table 3, PPTF is the second-best performing material: it is indeed the only one that showed a T_g_ value above room temperature, which means an amorphous glassy phase characterised by a low amount of free volume through which gases can diffuse. As for PPeTF and PHTF, both contained a rubbery amorphous phase at room temperature, coexisting with a mesophase and crystals. Consequently, their higher GTR values can be explained on the basis of the higher fraction of free volume due to T_g_ > RT and to the contemporary presence of both 3D and 2D domains causing higher disclination (channels created at the interface between the two ordered regions) density through which gas can easily diffuse. Conversely, there was no significant separation between the mesophase and amorphous regions, as they were characterised by a similar electron density.

Moreover, for all the samples under study, CO_2_ was more permeable than O_2_, in agreement with studies carried out in the literature on other similar polymeric systems [42,43] due to reduction of diffusivity with the decreasing of the permeant size (the values of molecular diameter for CO_2_ and O_2_ were 3.4 Å and 3.1 Å, respectively) [44].

The GTR_CO2_/GTR_O2_ ratio changed as a function of glycol subunit length; for PPTF (containing 3 methylene groups) it was close to 1, turning almost 2 in the case of PBTF, and 2.5 for PPeTF and PHTF (containing 5 and 6 methylene moieties, respectively). GTR_CO2_/GTR_O2_ ratio increments can be associated with a decrement of CO_2_ solubility in the polymer matrix occurring in samples containing longer glycol subunits.

If thiophene-based homopolymers are compared to their furan-based counterparts, some significant differences can be detected. First, within the thiophene family, the best-performing material was the one containing four carbon atoms in the glycol subunit, unlike what was found for the furan family where PPeF resulted in the best one [29,33]. It is worth noting both the samples were totally amorphous and with a T_g_ around RT; it can be supposed that this latter condition favoured the formation of a mesophase. It seems the optimal glycolic length to maximize mesophase formation was different for the two classes, probably because the two mesophases arose from different intermolecular interactions (π–π stacking in the case of poly(alkylene 2,5-thiophenedicarboxylate)s, π–π stacking and intermolecular hydrogen bonds for poly(alkylene 2,5-furandicarboxylate)s).

In addition, by comparing furan- and thiophene-based homopolymers containing a propylene segment, it can be noticed that GTR_PPTF_ < GTR_PPF_. Both were in the glassy state, but the former was semicrystalline, the latter amorphous. Mesophase formation was not favoured in both cases, the macromolecular chains being frozen in the glassy state. Considering the butylene moiety-containing polyesters, the results recorded evidence PBTF was more performant than PBF. That was probably due to the presence of a higher amount of mesophase, whose formation was favoured by an amorphous and partially mobile phase (PBF, on the contrary, was glassy and semicrystalline). The five methylene-containing polymers, PPeF and PPeTF, were both in the rubbery state at room temperature, with PPeTF containing also a crystal phase. Nevertheless, GTR_PPeF_ <<< GTR_PPeTF_ due to the very high mesophase fraction in the PPeF sample. Lastly, GTR_PHF_ < GTR_PHTF_ is explainable on the basis of a higher T_g_ and crystalline fraction for PHF.

Looking at the overall trend, by lengthening the glycol subunit in the poly(alkylene 2,5-thiophenedicarboxylate)s, we can observe a progressive worsening of barrier properties, except for PBTF due to the lowering of T_g_ values, unlike furan-based polyesters for which an even–odd trend was observed [29]. It can be assumed that for PPeTF and PHTF, despite the higher amount of mesophase, the worsening effect of both low T_g_ value and the high amount of disclinations between the mesophase and 3D domains prevailed.

The homopolymers under study were also compared to the fossil-based traditional materials currently employed in food packaging, as well as to other bio-based polyesters (see Figure 9B). Poly(alkylene 2,5-thiophenedicarboxylate)s turned out to be the more performant of the polyolefins and of PLA, PHB and PBS to both gases. Interestingly, PPTF and PBTF had even better barrier properties than PET and nylon.

## 4. Conclusions

A new family of fully bio-based homopolymers of 2,5-thiophenedicarboxylic acid were successfully synthesized via melt polycondensation, a green and solvent-free process, starting from dimethyl 2,5-thiophenedicarboxylate and glycols with different lengths.

All solid-state properties appeared to be strongly affected by glycol subunit length; the number of methylene groups present in the flexible aliphatic segment indeed impacted the macromolecular chain flexibility and crystallising ability, in turn, determining different glass transition temperature values and the development of different kinds and fractions of ordered phases (2D mesophase or 3D crystalline phase).

Through the analysis of the effect of glycol subunit length on sample microstructure, we could confirm some results previously reported:Mesophase formation is favoured in the case of macromolecular chains, which are mobile at room temperature but have reduced crystallising capability. If the glycol subunit is long enough, 3D crystalline phase formation appears prevalentMesophases and crystalline phases compete with each other, i.e., each one develops at the expense of the otherType, number, and amount of ordered phases have a huge impact on the final functional properties (mechanical and gas barrier)

As previously established for furan-based polyesters, the outstanding gas barrier properties are mainly ascribable to mesophase presence. However, due to the lower electronegativity of sulphur atoms with respect to oxygen ones, the 2D-ordered structure present in poly(alkylene 2,5-thiophenedicarboxylate)s could only originate from thiophene ring π–π stacking, different from furan-based polyesters where the mesophase also resulted from interchain hydrogen bonds involving furan rings.

This is the reason why the result of the comparison between the functional properties of a thiophene-based polyester and the furan-based analogue with the same length of the glycol subunit changed according to the length of this latter.

Of particular importance is the comparison in terms of functional properties between the best-performing polymer of the thiophene-based family, namely PBTF, and PPeF, the best polymer within the family of 2,5-furandicarboxylic acid. Both polyesters were characterised by outstanding barrier properties, even though some differences, directly related to the nature of interchain interactions responsible of the resulting mesophase, exist: the strong interchain hydrogen bonds established in PPeF in addition to van der Waals π–π interactions, also present among thiophene rings, explain the halving of the oxygen GTR value observed for the PPeF, whereas the lower dipole moment of the thiophene ring with respect to the furan one justified the doubling of the permselectivity ratio in the case of PBTF. In both cases, we are dealing with polymers with performances similar to EVOH, which is used as a high gas barrier material in multilayer films.

On the other hand, PBTF is more tough and ductile than PPeF, having an elastic modulus and a stress at break 5 and 3 times higher, respectively, than those of PPeF, the elongation at break being very high (>500%).

## Figures and Tables

**Figure 1 polymers-13-02460-f001:**
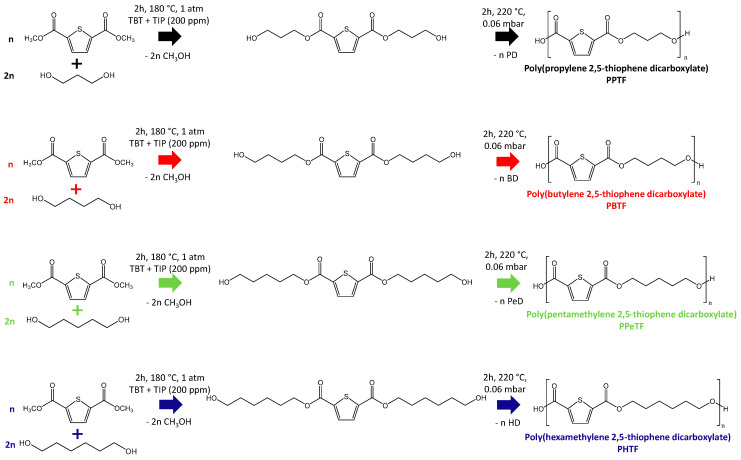
Schematic representation of the synthetic process together with the chemical structure of the homopolyesters under study: PPTF, PBTF, PPeTF and PHTF.

**Figure 2 polymers-13-02460-f002:**
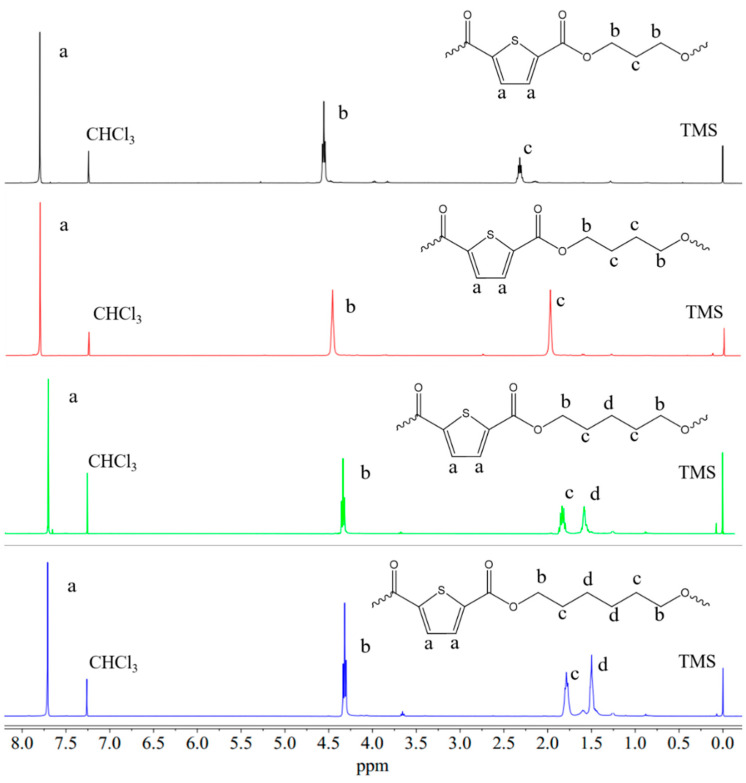
^1^H-NMR spectra of the homopolymers under study together with peaks’ assignment.

**Figure 3 polymers-13-02460-f003:**
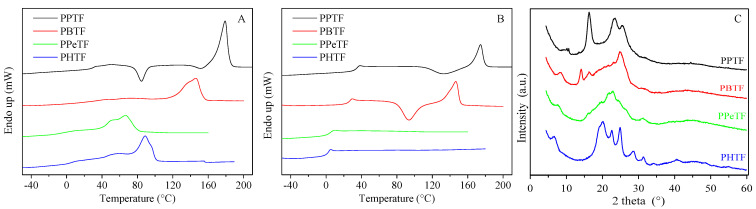
(**A**) I scan; (**B**) II scan DSC curves; (**C**) WAXS profiles of PPTF, PBTF, PPeTF and PHTF purified powders.

**Figure 4 polymers-13-02460-f004:**
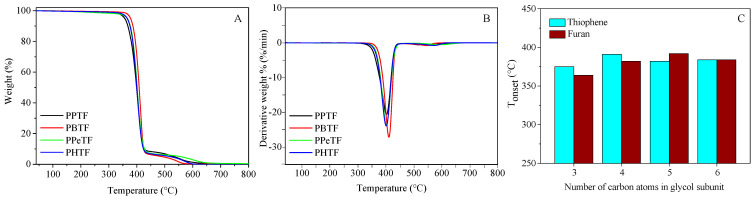
(**A**) TGA curves and (**B**) TGA derivatives of PPTF, PBTF, PPeTF and PHTF homopolymers; (**C**) comparison between T_onset_ of both thiophene- and furan-based homopolymers.

**Figure 5 polymers-13-02460-f005:**
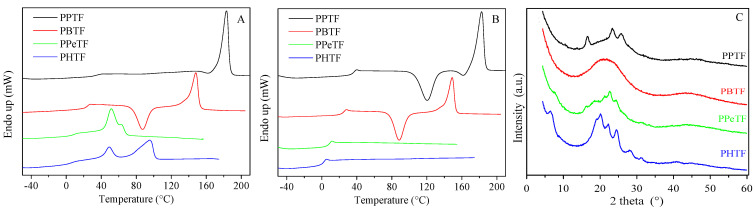
(**A**) I scan; (**B**) II scan DSC curves; (**C**) WAXS profiles of PPTF, PBTF, PPeTF and PHTF compression-moulded films.

**Figure 6 polymers-13-02460-f006:**
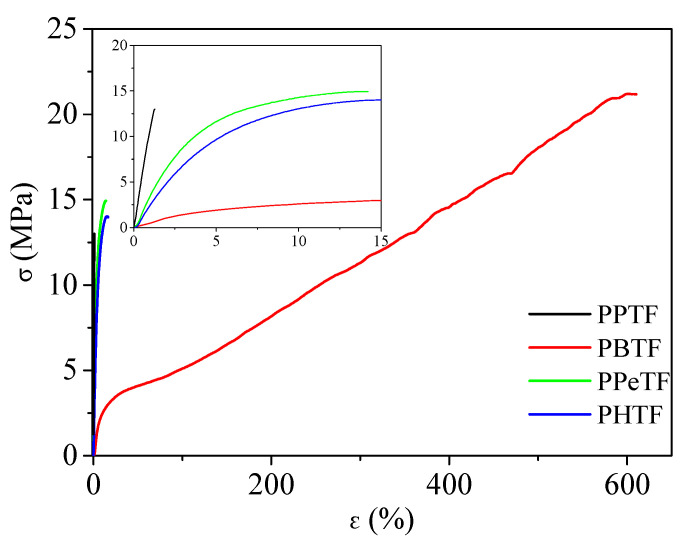
Stress–strain curves of PPTF, PBTF, PPeTF and PHTF films after 3 weeks of storage at room temperature.

**Figure 7 polymers-13-02460-f007:**
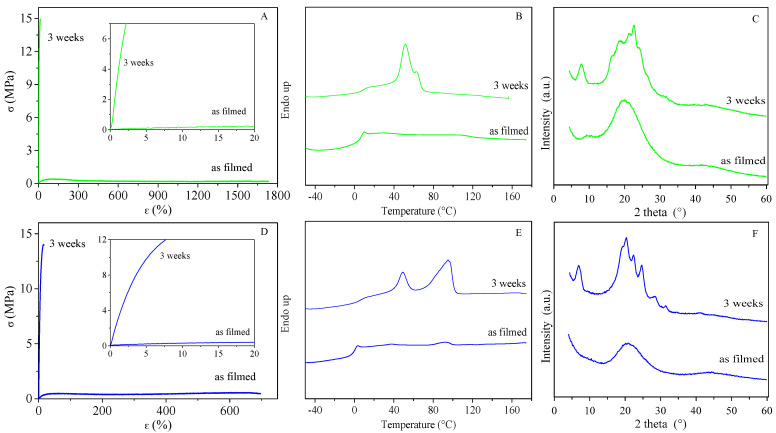
(**A**) Stress–strain curves; (**B**) DSC curves; (**C**) WAXS profiles of PPeTF films immediately after moulding and after 3 weeks of storage at room temperature. (**D**) Stress–strain curves; (**E**) DSC curves; (**F**) WAXS profiles of PHTF films immediately after moulding and after 3 weeks of storage at room temperature.

**Figure 8 polymers-13-02460-f008:**
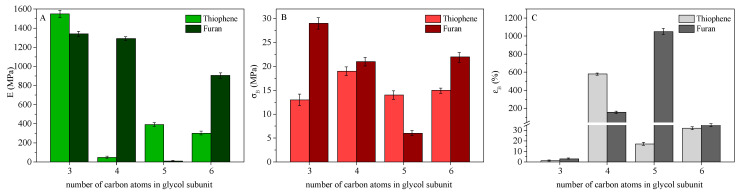
Comparison between mechanical characterisation data of thiophene-based homopolymers and their furan-based counterparts: (**A**) elastic modulus; (**B**) stress at break; (**C**) elongation at break.

**Figure 9 polymers-13-02460-f009:**
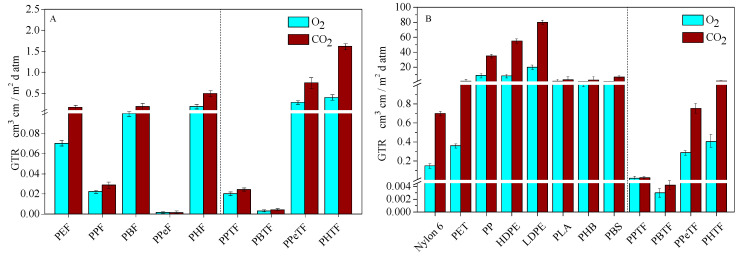
GTR values of O_2_ and CO_2_ through polymeric films (*T* = 23 °C) compared with those of (**A**) furan-based homopolymers [29,33]; (**B**) traditional plastics and principal bio-based polymers [8,18,45].

**Table 1 polymers-13-02460-t001:** ^1^H-NMR peak position and molecular characterisation data for the homopolymers under study.

	Peak Position (ppm)		M_n_ (Da)	D
	Diacid Subunit	Glycolic Subunit		
PPTF	7.83 (2H, s)	2.34 (2H, m) 4.60 (4H, t)	26,300	2.3
PBTF	7.83 (2H, s)	1.99 (4H, m) 4.48 (4H, t)	39,600	2.0
PPeTF	7.70 (2H, s)	1.58 (2H, m) 1.82 (4H, m) 4.34 (4H, t)	25,800	2.5
PHTF	7.70 (2H, s)	1.51 (4H, m) 1.81 (4H, m) 4.32 (4H, t)	32,400	2.3

**Table 2 polymers-13-02460-t002:** Thermal characterisation (TGA and DSC) data of the homopolymers under study in the form of compression-moulded films.

	T_onset_ °C	T_max_ °C	I Scan						II Scan					
			T_g_ °C	∆c_p_ J/g°C	T_cc_ °C	∆H_cc_ J/g	T_m_ °C	∆H_m_ J/g	T_g_ °C	∆c_p_ J/g°C	T_cc_ °C	∆H_cc_ J/g	T_m_ °C	∆H_m_ J/g
PPTF	375	402	36	0.175	-	-	183	44	36	0.323	120	36	183	37
PBTF	391	410	24	0.272	87	28	148	28	24	0.291	88	27	148	27
PPeTF	382	402	8	0.144	-	-	52/63	19/6	8	0.308	-	-	-	-
PHTF	384	402	7	0.008	-	-	49/95	10/23	2	0.289	-	-	-	-

**Table 3 polymers-13-02460-t003:** Mechanical and gas permeability characterisation data of the compression-moulded films.

	E (MPa)	σ_B_ (MPa)	ε_B_ (%)	O_2_-TR (cm^3^ cm m^−2^ d^−1^ atm^−1^)	CO_2_-TR (cm^3^ cm m^−2^ d^−1^ atm^−1^)
PPTF	1550 ± 125	13± 3	1.2 ± 0.3	0.021 ± 0.003	0.024 ± 0.002
PBTF	47 ± 3	19 ± 2	580 ± 60	0.003 ± 0.001	0.017 ± 0.001
PPeTF	1.9 ± 0.3 * 391 ± 26	0.5 ± 0.1 * 14 ± 2	1650 ± 110 * 17 ± 3	0.288 ± 0.035	0.753 ± 0.125
PHTF	3.5 ± 0.5 * 299 ± 22	0.5 ± 0.1 * 15 ± 1	674 ± 9 * 32 ± 3	0.404 ± 0.101	1.62 ± 0.110

* Mechanical data obtained immediately after film moulding.

## Data Availability

The data that support the findings of this study are available from the corresponding author upon reasonable request.

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
