# Peer review of "Poly(Alkylene 2,5-Thiophenedicarboxylate) Polyesters: A New Class of Bio-Based High-Performance Polymers for Sustainable Packaging"

_polymers, 2021, doi:10.3390/polym13152460_

Round 1

Reviewer 1 Report

Presented work is very interesting and valuable.

  • Section 2.2: Please, indicate the molar ratio of monomers during the synthesis of polymers. The amount of catalyst should be also indicated.
  • Section 2.4, Lines 145-146: "The so obtained films were cooled then in press to room temperature". How long does it take?
  • Section 2.4: Could you present the exemplary photograph of formed thin films?
  • HNMR and GPC spectra of polymers should be presented in manuscript
  • Table 3: "Mechanical data obtained immediately after film molding" - After cooling to room temperature?
  • Figure 7: The presented average values should be supported by standard deviations
  • Please, present DTG curves and discuss the mechanism of thermal decomposition
  • I suggest to determine a melt flow index of obtained polymers
  • How did you estimate the storage time (before tests) of formed films? Did you measure mechanical or thermal properties during the storage?
  • The crystallization kinetics (including different temperature) will be valuable completion of work.

Author Response

First of all, we would like to thank the Reviewers for their time and very valuable suggestions that helped us improving the final version of our article.

Please, find in the following the point to point response to Reviewers’ reports.

Reviewer 1

Section 2.2: Please, indicate the molar ratio of monomers during the synthesis of polymers. The amount of catalyst should be also indicated.

We have modified the text as kindly suggested.

Section 2.4, Lines 145-146: "The so obtained films were cooled then in press to room temperature". How long does it take?

The samples were ballistically cooled down to room temperature, the process taking 15 minutes.

We have modified the text accordingly.

Section 2.4: Could you present the exemplary photograph of formed thin films?

We have reported the pictures of the compression molded films in the SI.

HNMR and GPC spectra of polymers should be presented in manuscript.

We have moved the NMR spectra to the main article and added the GPC curves in the SI.

Table 3: "Mechanical data obtained immediately after film molding" - After cooling to room temperature?

Yes, the samples were subjected to mechanical just after cooling from the melt to room temperature.

Figure 7: The presented average values should be supported by standard deviations.

We have added the standard deviations as suggested.

Please, present DTG curves and discuss the mechanism of thermal decomposition.

We have added DTG curves and implemented the discussion.

I suggest to determine a melt flow index of obtained polymers.

Unfortunately, we cannot perform this measurement since we do not have the instrument required.

How did you estimate the storage time (before tests) of formed films? Did you measure mechanical or thermal properties during the storage?

We measured the thermal properties.

The crystallization kinetics (including different temperature) will be valuable completion of work.

We thank the reviewer for the suggestion, this point deserves for sure to be carried out through microscopical as well as calorimetric techniques in a further study.

Reviewer 2 Report

The authors have reported thiophene dicarboxylate-based polyesters which have recently gained a lot of attention because of great barrier properties that can compete with existing commercial materials. The work is of great interest to the sustainable chemistry community and is presented well. However, I suggest some minor revisions. 

  1. Line 22: The sentence should be 'responsible for the different....".
  2. Line 66: FDCA should be written in brackets next to the full form.
  3. Section 2.2: Synthesis scheme must be drawn clearly showing the reaction along with the conditions and all chemical structures of all the monomers and polymers. This is very important to show as the whole paper is explained based on the chemical structures of the polymers. 
  4. Line 143: Please mention the Model number of the instrument and the country. 
  5. Line 143 to 144: Please mention the melting point of the polymers and the hot pressing temperatures clearly, instead of writing "melted at a temperature 30° higher than their melting temperature".
  6. Line 170: Write the exact temperature window of the DSC scan instead of writing "Tm + 30deg". 
  7. Figure 2A and 2B: Units should be written on the Y-axis along with the paprameter (Heat flow)
  8. Same in Figure 4A and 4B.
  9. Units are missing in the y-axis of Figure 8. Also, please include the standard deviation in the graph. 
    Also, the graph is not placed at proper angles because of which it is difficult to read the values in the figure. 2D graph is suggested for better clarity.   
  10. English language of the paper must be improved. 

Author Response

First of all, we would like to thank the Reviewers for their time and very valuable suggestions that helped us improving the final version of our article.

Please, find in the following the point to point response to Reviewers’ reports.

Reviewer 2

The authors have reported thiophene dicarboxylate-based polyesters which have recently gained a lot of attention because of great barrier properties that can compete with existing commercial materials. The work is of great interest to the sustainable chemistry community and is presented well. However, I suggest some minor revisions. 

  1. Line 22: The sentence should be 'responsible for the different....”.

We have modified the sentence as suggested.

  1. Line 66: FDCA should be written in brackets next to the full form.

We have modified the text accordingly.

  1. Section 2.2: Synthesis scheme must be drawn clearly showing the reaction along with the conditions and all chemical structures of all the monomers and polymers. This is very important to show as the whole paper is explained based on the chemical structures of the polymers.

We have added a schematic representation of the syntheses together with the chemical formulas of all the polymers prepared.

  1. Line 143: Please mention the Model number of the instrument and the country.

We have added this information.

  1. Line 143 to 144: Please mention the melting point of the polymers and the hot pressing temperatures clearly, instead of writing "melted at a temperature 30° higher than their melting temperature”.

We have modified the text as suggested.

  1. Line 170: Write the exact temperature window of the DSC scan instead of writing "Tm + 30deg".

We have modified the text as suggested.

  1. Figure 2A and 2B: Units should be written on the Y-axis along with the parameter (Heat flow)

We have modified the text as suggested.

  1. Same in Figure 4A and 4B.

We have modified the text as suggested.

  1. Units are missing in the y-axis of Figure 8. Also, please include the standard deviation in the graph. 
    Also, the graph is not placed at proper angles because of which it is difficult to read the values in the figure. 2D graph is suggested for better clarity.

We have replaced the 3D graph with a 2D one, also adding the y-axis units.

  1. English language of the paper must be improved.

We have revised the manuscript as suggested.

Round 2

Reviewer 1 Report

Manuscript was improved in the terms of reviewers comments, so in my opinion work can be accepted in present form. Thank you very much for your responses and introduced completions.